# HBO: Hierarchical Balancing Optimization for Fine-Tuning Large Language Models

**Weixuan Wang**[1†]   **Minghao Wu**[2†]   **Barry Haddow**[1,3]   **Alexandra Birch**[1,3]

[1]School of Informatics, University of Edinburgh
[2]Tongyi Lab, Alibaba Group
[3]Aveni
{weixuan.wang, bhaddow, a.birch}@ed.ac.uk
minghao.wu@alibaba-inc.com

## Abstract

Fine-tuning large language models (LLMs) on a mixture of diverse datasets poses challenges due to data imbalance and heterogeneity. Existing methods often address these issues across datasets (globally) but overlook the imbalance and heterogeneity within individual datasets (locally), which limits their effectiveness. We introduce ***Hierarchical Balancing Optimization (HBO)***, a novel method that enables LLMs to autonomously adjust data allocation during fine-tuning both across datasets (globally) and within each individual dataset (locally). *HBO* employs a bilevel optimization strategy with two types of actors: a Global Actor, which balances data sampling across different subsets of the training mixture, and several Local Actors, which optimizes data usage within each subset based on difficulty levels. These actors are guided by reward functions derived from the LLM's training state, which measure learning progress and relative performance improvement. We evaluate *HBO* on three LLM backbones across nine diverse tasks in multilingual and multitask setups. Results show that *HBO* consistently outperforms existing baselines, achieving significant accuracy gains. Our in-depth analysis further demonstrates that both the global actor and local actors of *HBO* effectively adjust data usage during fine-tuning. *HBO* provides a comprehensive solution to the challenges of data imbalance and heterogeneity in LLM fine-tuning, enabling more effective training across diverse datasets. In addition, we release the code to foster research along this line.[1]

## 1 Introduction

Large language models (LLMs) have demonstrated remarkable capabilities in understanding, reasoning, and generating answers across diverse tasks (Reid et al., 2024; DeepSeek-AI et al., 2025; OpenAI, 2024a;c). One of the key factors contributing to the success of LLMs is the dataset mixture used during their fine-tuning process (Taori et al., 2023; Yang et al., 2024b; Muennighoff et al., 2023; Chung et al., 2024a). These dataset mixtures typically encompass a diverse range of tasks, domains, and languages, ensuring that the models can perform well across a wide variety of applications (Yang et al., 2024a; Dubey et al., 2024; Team et al., 2025; Martins et al., 2024; Aryabumi et al., 2024).

A critical challenge in fine-tuning LLMs is managing data **imbalance** and **heterogeneity**, which arise from the diverse nature of tasks and datasets used for fine-tuning. Data imbalance refers to the uneven distribution of examples across different tasks, domains, or languages (Liu et al., 2020; Kamalov & Denisov, 2020; Pouyanfar et al., 2018; Wang et al., 2019), while data heterogeneity encompasses variations in the characteristics of the data, such as quality and difficulty (Liu et al., 2024; Hendrycks et al., 2021b; Li et al., 2024b; Albalak et al., 2023). Addressing these factors is crucial for achieving optimal performance, as imbalanced or heterogeneous data can lead to overfitting on certain tasks and underperformance on others (Zhao et al., 2023; Li et al., 2024b). Recent research highlights the

---

[†] Equal contribution.
[1]https://github.com/weixuan-wang123/HBO

importance of strategically balancing data from various sources to mitigate these issues (Wei et al., 2021; Iyer et al., 2022; Wu et al., 2024b). However, existing methods often assume that datasets are internally balanced and homogeneous, which may not hold true in practice, as the examples from the same sources may also exhibit different characteristics (Schwartz & Stanovsky, 2022; He & Garcia, 2009). This limitation underscores the need for more effective strategies that can manage imbalance and heterogeneity both **globally (across datasets)** and **locally (within datasets)**. Developing such strategies is challenging, as finding the optimal allocation of data usage across these dimensions requires substantial efforts and resources. This raises a natural research question:

> *Can LLMs determine optimal data usage strategies by themselves to address imbalance and heterogeneity?*

To address this research question, we propose a novel framework, ***Hierarchical Balancing Optimization (HBO)***, that enables LLMs to autonomously adjust their data allocation both globally and locally based on their current training state. Our method employs a bilevel optimization framework (Colson et al., 2007), where the outer problem minimizes the objective function of training the LLM over a mixture of training datasets, and the inner problem adjusts the sampling probabilities both **globally (across datasets)** and **locally (within datasets)**. To achieve this optimization, we introduce two types of actors: **Global Actor** and **Local Actor**. The global actor is responsible for balancing data allocation across the subsets of the training data mixture, while the local actor for each individual subset optimizes data usage within the subset. Specifically, we categorize the examples in each subset into four groups based on their difficulty levels. To guide the global and local actors, we define two reward functions based on the training state of the LLM. The global reward is computed as the $L_2$ norm of the gradients, which reflects the current learning progress of the model. The local reward is defined as the ratio of the perplexities given by the fine-tuned LLM and the original LLM, capturing the relative improvement in model performance on specific groups. By integrating these components, our *HBO* framework effectively optimizes the training process of LLMs.

To validate the effectiveness of *HBO*, we conduct extensive experiments using three model backbones: Llama-3.1-8B, Qwen2.5-7B, and EuroLLM-9B. These experiments span two setups, covering a total of nine tasks: a Multilingual setting (MMLU, XCOPA, XStoryCloze, XNLI, and MGSM) and a Multitask setting (MMLU, MultiFin-EN, GSM8K, and MedMCQA). Our results demonstrate that *HBO* consistently outperforms existing sampling strategies, achieving substantial improvements in model performance.

Our contributions in this work can be summarized as follows:

- We propose *HBO*, a novel hierarchical dynamic data sampling method that enables LLMs to autonomously address data imbalance and heterogeneity during fine-tuning. By leveraging a bilevel optimization framework with two types of actors, *HBO* dynamically adjusts sampling probabilities both globally and locally (see Section 3).
- We demonstrate the broad applicability and effectiveness of *HBO* across multiple LLM backbones and tasks. Through extensive experiments with three model backbones over nine diverse tasks, *HBO* consistently outperforms existing sampling strategies, achieving substantial accuracy improvements. Visualizations of the sampling probabilities reveal that *HBO* dynamically adjusts these probabilities in a fascinating cyclical pattern, highlighting its ability to adaptively focus on areas that enhance model learning (see Section 4.2).
- We conduct extensive analyses to investigate the contributions of the global and local actors, the robustness of *HBO* to varying sampling priors, the impact of data volume, and more. Additionally, we highlight the critical role of incorporating easy examples, which are often discarded during fine-tuning, in boosting model performance (see Section 5).

## 2 RELATED WORK

**Fine-Tuning with Multi-Datasets**   Recent advances in LLMs have shown that utilizing diverse datasets for both pre-training and fine-tuning is crucial for developing robust, generalized models (Team et al., 2024; Anthropic, 2024; OpenAI, 2024b;c). Multi-task learning, which fine-tunes models on multiple tasks or languages simultaneously, leverages shared knowledge and improves overall performance (Crawshaw, 2020; Zhang et al., 2022; Le Scao et al., 2023; Aryabumi et al., 2024). Despite these successes, critical challenges remain, particularly in balancing diverse task objectives

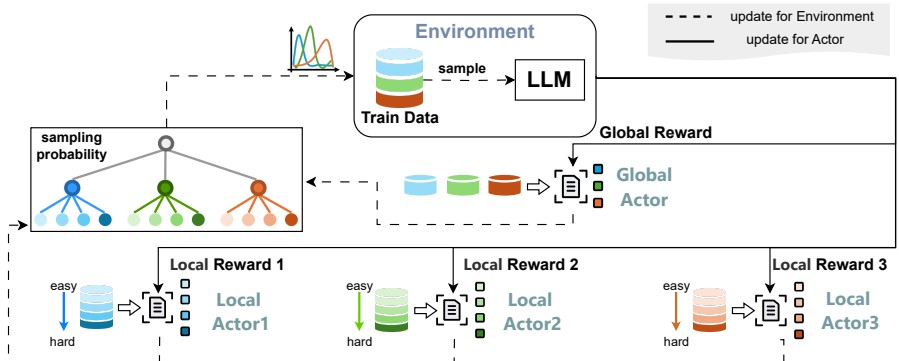

Figure 1: The bilevel optimization framework of **HBO**. **Global Actor** and **Local Actors** jointly adjust data sampling probabilities both globally (across datasets) and locally (within datasets) to optimize the LLM parameters. Based on the LLM's training state, the global reward and the local reward are computed to guide the optimization of the global and local actors, respectively.

(Chen et al., 2018; Kendall et al., 2018; Yu et al., 2020; Wang et al., 2020c; Khalifa et al., 2024) and enhancing cross-lingual transferability (Kew et al., 2024; Chen et al., 2024; Shaham et al., 2024).

**Data Balancing**   Recent studies emphasize strategies to ensure that training datasets are not only diverse but also reflective of various domain-specific characteristics (Wei et al., 2021; Iyer et al., 2022; Chung et al., 2024b; Liu et al., 2020; Kamalov & Denisov, 2020). Prior static balancing approaches apply fixed sampling probabilities throughout training (Arivazhagan et al., 2019; Conneau et al., 2020). In contrast, dynamic balancing methods adapt these probabilities over time (Wang et al., 2020b; Wu et al., 2021; Zhu et al., 2024; Wang et al., 2019), often by incorporating another scorer network, to more effectively manage the evolving learning process (Wang et al., 2020b;a; Wu et al., 2024a). However, most of these methods primarily address differences between datasets (global balancing) while often neglecting the internal diversity found within a single dataset (local balancing).

**Ours**   Our work introduces *HBO* that addresses data imbalances and heterogeneity at both global and local levels. Unlike prior methods that focus solely on global balancing (Wang et al., 2020b; Wu et al., 2024a), our approach simultaneously optimizes sampling across datasets and within individual datasets, ensuring more effective utilization of heterogeneous training data.

## 3 METHODOLOGY

### 3.1 PRELIMINARIES

**Supervised Fine-Tuning**   Supervised fine-tuning (SFT) is a process that enables LLMs to follow and respond to human instructions. During SFT, an LLM with parameters $\boldsymbol{\theta}$ is trained on instruction-response pairs to improve its ability to generate helpful, accurate responses to user queries. For a single training dataset with $M_1$ examples $\mathcal{D}^1 = \{(\boldsymbol{x}_k, \boldsymbol{y}_k)\}_{k=1}^{M_1}$, $\boldsymbol{x}_k$ is the instruction (input prompt), and $\boldsymbol{y}_k$ is the desired response, the SFT objective minimizes the negative log-likelihood:

$$\mathcal{L}_s(\mathcal{D}^1; \boldsymbol{\theta}) = -\sum_{k=1}^{M_1} \log p(\boldsymbol{y}_k | \boldsymbol{x}_k; \boldsymbol{\theta}) \tag{1}$$

In practical applications, LLMs are often fine-tuned on multiple diverse datasets $\mathcal{D} = \{\mathcal{D}^i\}_{i=1}^{N}$, where each dataset $\mathcal{D}^i = \{(\boldsymbol{x}_k^i, \boldsymbol{y}_k^i)\}_{k=1}^{M_i}$ contains $M_i$ examples. In this multi-dataset setting, the naive approach combines the losses across all datasets:

$$\mathcal{L}(\mathcal{D}; \boldsymbol{\theta}) = \sum_{i=1}^{N} \mathcal{L}_s(\mathcal{D}^i; \boldsymbol{\theta}) \tag{2}$$

---

**Algorithm 1:** *Hierarchical Balancing Optimization*

---

**Input** : $\mathcal{D} = \{\{\{(\boldsymbol{x}_k^{i,j}, \boldsymbol{y}_k^{i,j})\}_{k=1}^{Q_{i,j}}\}_{j=1}^{M_i}\}_{i=1}^N$, a training set organized into $N$ subsets, where each subset $i$ contains $M_i$ groups, and each group $j$ holds $Q_{i,j}$ pairs consisting of instruction $\boldsymbol{x}_k^{i,j}$ and their response $\boldsymbol{y}_k^{i,j}$; $F_{\text{global}}$ and $\gamma_{\text{global}}$, update frequency and learning rate for $\boldsymbol{\psi}_{\text{global}}$; $F_{\text{local}}$ and $\gamma_{\text{local}}$, update frequency and learning rate for $\boldsymbol{\psi}_{\text{local}}$; $T$, total training steps; $\alpha$, learning rate for $\boldsymbol{\theta}$;

**Output :** The converged model $\boldsymbol{\theta}$;

1 Initialize $p_{\boldsymbol{\psi}_{\text{global}}}(N)$ and $p_{\boldsymbol{\psi}_{\text{local}}}(M_i)$ using Equation 3 with $\tau = 1$ ;

2 **for** $t = 0$ *to* $T$ **do**

3     $\tilde{i} \sim p_{\boldsymbol{\psi}_{\text{global}}}(N)$;

4     $\tilde{j} \sim p_{\boldsymbol{\psi}_{\text{local}}^{\tilde{i}}}(M_{\tilde{i}})$ ;

5     Sample batch $(\boldsymbol{x}, \boldsymbol{y}) \sim \mathcal{D}^{\tilde{i},\tilde{j}}$ ;

6     $\boldsymbol{\theta} \leftarrow \boldsymbol{\theta} - \alpha \cdot \nabla_{\boldsymbol{\theta}} \mathcal{L}(\boldsymbol{y}|\boldsymbol{x}; \boldsymbol{\theta})$;

7     **if** $t \% F_{global} == 0$ **then**

8        **for** $i = 1$ *to* $N$ **do**

9           $(\boldsymbol{x}', \boldsymbol{y}') \sim \mathcal{D}^i$;

10           Compute reward $\mathcal{R}_{\text{global}}(i)$ for $\mathcal{D}^i$ as in Section 3.3 ;

11        **end**

12        $\boldsymbol{\psi}_{\text{global}} \leftarrow \boldsymbol{\psi}_{\text{global}} + \sum_{i=1}^N \gamma_{\text{global}} \cdot \mathcal{R}_{\text{global}}(i) \cdot \nabla_{\boldsymbol{\psi}_{\text{global}}} \log p_{\boldsymbol{\psi}_{\text{global}}}(i)$ ;

13     **end**

14     **if** $t \% F_{local} == 0$ **then**

15        **for** $i = 1$ *to* $N$ **do**

16           **for** $j = 1$ *to* $M_i$ **do**

17              $(\boldsymbol{x}', \boldsymbol{y}') \sim \mathcal{D}^{i,j}$;

18              Compute reward $\mathcal{R}_{\text{local}}(i, j)$ for $\mathcal{D}^{i,j}$ as in Section 3.3 ;

19           **end**

20           $\boldsymbol{\psi}_{\text{local}} \leftarrow \boldsymbol{\psi}_{\text{local}} + \sum_{j=1}^{M_i} \gamma_{\text{local}} \cdot \mathcal{R}_{\text{local}}(i, j) \cdot \nabla_{\boldsymbol{\psi}_{\text{local}}} \log p_{\boldsymbol{\psi}_{\text{local}}}(i, j)$ ;

21        **end**

22     **end**

23 **end**

---

**Static Balancing** When fine-tuning on multiple datasets of varying sizes, simply merging them underrepresent ones. Static balancing addresses this issue by adjusting each dataset's sampling probability with a temperature parameter $\tau$ (Arivazhagan et al., 2019; Conneau et al., 2020). The base sampling probability of the $i$-th dataset is : $q(i) = \frac{M_i}{\sum_{n=1}^N M_n}$, then adjusted using temperature $\tau$ as:

$$q_\tau(i) = \frac{q(i)^{1/\tau}}{\sum_{n=1}^N q(n)^{1/\tau}} \tag{3}$$

The temperature parameter $\tau$ provides flexible control over dataset representation: $\tau = 1$ yields proportional sampling based on dataset sizes (equivalent to the naive approach in Equation 2), while as $\tau$ approaches $\infty$, sampling becomes increasingly uniform across datasets regardless of their original sizes. With this temperature-adjusted sampling, the training objective becomes:

$$\mathcal{L}(\mathcal{D}; \boldsymbol{\theta}, q_\tau) = \mathbb{E}_{i \sim q_\tau} \left[ \mathcal{L}_s(\mathcal{D}^i; \boldsymbol{\theta}) \right] \tag{4}$$

## 3.2 Hierarchical Balancing Optimization

We introduce *Hierarchical Balancing Optimization (HBO)*, a hierarchical dynamic data sampling framework designed to address both global (across datasets) and local (within datasets) heterogeneity. Our approach leverages bilevel optimization to jointly optimize the LLM parameters $\boldsymbol{\theta}$ and two types of actors: the **Global Actor** $\boldsymbol{\psi}_{\text{global}}$ and the **Local Actor** $\boldsymbol{\psi}_{\text{local}}$.[2] As shown in Figure 1, the LLM $\boldsymbol{\theta}$ and the training datasets $\mathcal{D}$ form the *environment*, while the actors $\boldsymbol{\psi}_{\text{global}}$ and $\boldsymbol{\psi}_{\text{local}}$ act as *agents* that dynamically adjust sampling probabilities. The global actor $\boldsymbol{\psi}_{\text{global}}$ balances the contributions of

---

[2] The $\boldsymbol{\psi}_{\text{local}}$ represents the collection of the local actors for each individual dataset and the $\boldsymbol{\psi}_{\text{local}}^i$ indicates the local actor for the $i$-th dataset.

different datasets, ensuring fair representation across datasets of varying sizes. Meanwhile, the local actors $\psi_{\text{local}}$ adjusts sampling probabilities within each dataset, accounting for internal heterogeneity.

Our framework is formulated as a bilevel optimization problem (Colson et al., 2007), where the solution to the inner problem constrains the outer problem. In our case, the outer optimization minimizes the objective $\mathcal{J}(\mathcal{D}; \boldsymbol{\theta})$, which evaluates the LLM's performance on the training datasets. The inner optimization adjusts the sampling probabilities $p_{\psi_{\text{global}}}(N)$ and $p_{\psi_{\text{local}}^i}(M_i)$ based on the LLM's performance. This results in the following bilevel optimization formulation:

$$(\psi_{\text{global}}, \psi_{\text{local}}) = \underset{\psi_{\text{global}}, \psi_{\text{local}}}{\arg\min} \ \mathcal{J}(\mathcal{D}; \boldsymbol{\theta}(\psi_{\text{global}}, \psi_{\text{local}})), \text{ where}$$

$$\boldsymbol{\theta}(\psi_{\text{global}}, \psi_{\text{local}}) = \underset{\boldsymbol{\theta}}{\arg\min} \ \mathbb{E}_{i \sim p_{\psi_{\text{global}}}(N)} \left[ \mathbb{E}_{j \sim p_{\psi_{\text{local}}^i}(M_i)}[\mathcal{L}((\boldsymbol{x}^{i,j}, \boldsymbol{y}^{i,j}); \boldsymbol{\theta})] \right] \quad (5)$$

This hierarchical optimization framework ensures that both global and local sampling probabilities are dynamically adjusted to maximize the LLM's performance, enabling more effective utilization of heterogeneous training data.

We present the detailed implementation of *HBO* in Algorithm 1. We organize the training data into a hierarchical structure with $N$ subsets, where each subset $i$ contains $M_i$ groups. The algorithm begins by initializing both global and local sampling distributions with temperature $\tau = 1$. During each training iteration, we first sample a subset $\tilde{i}$ according to the global distribution $p_{\psi_{\text{global}}}(N)$, then select a group $\tilde{j}$ based on the local distribution $p_{\psi_{\text{local}}^{\tilde{i}}}(M_{\tilde{i}})$. After updating the model parameters $\boldsymbol{\theta}$ using the sampled batch, we periodically update both actors. The global actor $\psi_{\text{global}}$ is updated every $F_{\text{global}}$ steps by evaluating rewards $\mathcal{R}_{\text{global}}(i)$ for each subset, while the local actors $\psi_{\text{local}}$ are updated every $F_{\text{local}}$ steps by computing rewards $\mathcal{R}_{\text{local}}(i, j)$ for each group. These updates follow a policy gradient approach, where the gradient of the log probability is scaled by the corresponding reward. This optimization strategy ensures that sampling probabilities at both hierarchical levels are continuously refined to maximize model performance.

A critical issue in Algorithm 1 is that Equation 5 is not directly differentiable with respect to the actors $\psi_{\text{global}}$ and $\psi_{\text{local}}$. We address this using the Reinforce algorithm (Williams, 1992), a policy gradient method that enables gradient-based optimization of non-differentiable rewards. In *HBO*, we compute rewards based on the model's performance on sampled training data, then update the policy parameters to maximize expected rewards. The updates for $\psi_{\text{global}}$ and $\psi_{\text{local}}$ take the unified form:

$$\psi \leftarrow \psi + \gamma \cdot \mathcal{R} \cdot \nabla_{\psi} \log p_{\psi}(\cdot) \quad (6)$$

where $\psi$ represents either $\psi_{\text{global}}$ or $\psi_{\text{local}}$, $\gamma$ is the learning rate ($\gamma_{\text{global}}$ or $\gamma_{\text{local}}$), $\mathcal{R}$ is the computed reward ($\mathcal{R}_{\text{global}}(i)$ or $\mathcal{R}_{\text{local}}(i, j)$), and $p_{\psi}(\cdot)$ is the probability of selecting a particular group or subset.

**Implementation Details** In *HBO*, both the global actor $\psi_{\text{global}}$ and local actors $\psi_{\text{local}}$ are implemented as 2-layer fully connected network. Each actor takes as input a feature vector representing the corresponding sampling unit (subset or group). This lightweight architecture is sufficient because the actors only need to model relatively simple distributions over the hierarchical training data structure. It is important to note that these actors are used exclusively for optimizing data sampling during the fine-tuning process, incurring about a 15% additional training overhead in total training runtime compared with static balanced sampling. They are entirely separate from any reward models typically employed in Reinforcement Learning with Human Feedback (RLHF). Furthermore, we employ the SuperFiltering metric (Li et al., 2024a) to evenly divide each subset into four groups based on task difficulty. Group 1 contains the easiest examples, while Group 4 contains the hardest examples.

### 3.3 A Tale of Two Rewards

In *HBO*, we introduce two distinct reward functions to guide the optimization of the global and local actors. The global reward $\mathcal{R}_{\text{global}}(i)$ evaluates the performance of the LLM on a specific subset $i$, while the local reward $\mathcal{R}_{\text{local}}(i, j)$ assesses the performance on a specific group $j$ within subset $i$. These rewards are designed to capture different aspects of model performance and are crucial for effective sampling in heterogeneous datasets.

**Global Reward** Recent work demonstrates that the $L_2$ norm of the gradients decreases as the model gradually learns (Chen et al., 2018), suggesting that the $L_2$ norm of the gradients is an ideal

signal of learning dynamics of the LLM across various datasets. Based on this insight, given a random batch $B_i = \{(\boldsymbol{x}^i, \boldsymbol{y}^i)\}$ uniformly sampled from subset $i$, we define our global reward $\mathcal{R}_{\text{global}}(i)$ as the $L_2$ norm of the gradients computed on subset $i$:

$$\mathcal{R}_{\text{global}}(i) = \|\nabla_{\boldsymbol{\theta}} \mathcal{L}(B_i; \boldsymbol{\theta})\|_2 \tag{7}$$

This reward mechanism prioritizes datasets with larger gradient norms, effectively allocating more training resources to subsets where the model has more to learn. As the model becomes more proficient on a particular subset, the corresponding gradient norm naturally decreases, causing the global actor $\boldsymbol{\psi}_{\text{global}}$ to gradually shift focus to other subsets where improvement is still needed.

**Local Reward** The local heterogeneity typically stems from the varying difficulty of examples within each subset. To address this, we design a local reward mechanism that effectively prioritizes examples based on their learning progress. Inspired by Wu et al. (2024a), we define the local reward $\mathcal{R}_{\text{local}}(i, j)$ as the ratio between the current perplexity and the initial perplexity of the model on examples from group $j$ within subset $i$. This ratio serves as an indicator of learning progress, where a higher value suggests the model has made less progress on these examples relative to its starting point. Specifically, given a random batch $B_{i,j} = \{(\boldsymbol{x}_k^{i,j}, \boldsymbol{y}_k^{i,j})\}_{k=1}^K$ sampled from group $j$ of subset $i$, $\mathcal{R}_{\text{local}}(i, j)$ is defined as:[3]

$$\mathcal{R}_{\text{local}}(i, j) = \frac{1}{K} \sum_{k=1}^K \frac{\text{PPL}(\boldsymbol{y}_k; \boldsymbol{x}_k, \boldsymbol{\theta})}{\text{PPL}(\boldsymbol{y}_k; \boldsymbol{x}_k, \boldsymbol{\theta}_0)}, \text{ where}$$

$$\text{PPL}(\boldsymbol{y}_k; \boldsymbol{x}_k, \boldsymbol{\theta}) = \exp\left(-\frac{1}{|\boldsymbol{y}_k|} \sum_{l=1}^{|\boldsymbol{y}_k|} \log p(\boldsymbol{y}_{k,l} | \boldsymbol{x}_k, \boldsymbol{y}_{k,<l}; \boldsymbol{\theta})\right) \tag{8}$$

Here, $\boldsymbol{\theta}$ denotes the current model parameters, $\boldsymbol{\theta}_0$ represents the initial model parameters, $|\boldsymbol{y}_k|$ is the length of the response $\boldsymbol{y}_k$, and $K$ is the batch size. For examples where the model improves quickly, $\mathcal{R}_{\text{local}}(i, j)$ decreases, reducing their sampling probability. Conversely, examples where improvement is lower maintain higher rewards.

## 4 EXPERIMENT

### 4.1 EXPERIMENTAL SETUP

**Multitask Setup** We construct our training mixture from four distinct domains: general, math, financial, and medical. The `General` dataset is collected from the English part of the `WildChat` (Zhao et al., 2024) and `LMSYS-Chat-1M` (Zheng et al., 2024), resulting in a total of 1,196K examples. The `Math` dataset is derived from the `MetaMathQA` (Yu et al., 2024), containing 393K examples. The `Finance` dataset is obtained from the AQ22 [4], with 256K examples. Finally, the `Medical` dataset is sourced from the `UltraMedical` (Zhang et al., 2024), comprising 409K examples. Accordingly, we conduct zero-shot evaluations on the following testsets: MMLU (Hendrycks et al., 2021a) for general, GSM8K (Cobbe et al., 2021) for math, `MultiFin-EN` (Jørgensen et al., 2023) for finance, and MedMCQA (Pal et al., 2022) for medical. Due to computational constraints, we sample 10% of total examples from each training dataset. Task performance is evaluated using the *accuracy* metric given by (Gao et al., 2024), and the overall performance is reported as the macro-average across all tasks ($\mu_{\text{MT}}$).

**Multilingual Setup** We fine-tune multilingual LLMs using a combination of `Aya Dataset` (Singh et al., 2024) and `WildChat` (Zhao et al., 2024), covering eight languages: English (273K), Arabic (12K), German (6K), Spanish (15K), Hindi (1K), Russian (49K), Swahili (578), and Chinese (102K). Zero-shot evaluations are conducted on diverse downstream tasks, including MMMLU (Hendrycks et al., 2021a), XCOPA (Ponti et al., 2020), XStoryCloze (Lin et al., 2021), XNLI (Conneau et al., 2018), and MGSM (Shi et al., 2022). To address computational constraints, we sample 20% of the examples from each training dataset. Task performance is measured as macro-average *accuracy* across languages (Gao et al., 2024). The overall performance is reported as the macro-average across all tasks ($\mu_{\text{ML}}$).

---

[3]We omit the superscript $i, j$ in the notation for brevity.

[4]https://huggingface.co/datasets/DeividasM/financial-instruction-aq22

Table 1: Main results given by EuroLLM-9B, Llama-3.1-8B and Qwen2.5-7B on the multilingual and multitask settings. XSC and M.Fin are the XStoryCloze and MultiFin-EN datasets. The best results and second-best results are highlighted. † indicates statistical significance against the best baseline at $p < 0.05$ in terms of $\mu_{\text{ML}}$ and $\mu_{\text{MT}}$, following (Koehn, 2004).

| | Multilingual | | | | | | Multitask | | | | |
|---|---|---|---|---|---|---|---|---|---|---|---|
| | $\mu_{\text{ML}}$ | MMMLU | MGSM | XCOPA | XSC | XNLI | $\mu_{\text{MT}}$ | MMLU | M.Fin | GSM8K | MedMCQA |
| **EuroLLM-9B** | | | | | | | | | | | |
| **Heuristic** | | | | | | | | | | | |
| Prop. | 48.50 | 45.29 | 21.27 | 65.00 | 66.32 | **44.59** | 49.10 | 55.13 | 53.48 | 46.70 | 41.09 |
| Temp. | 47.32 | 43.38 | 19.07 | 65.40 | 66.09 | 42.67 | 48.43 | 55.36 | 53.11 | 43.14 | 42.12 |
| Uni. | 47.49 | 42.64 | 21.13 | 65.20 | 66.25 | 42.23 | 48.07 | 55.38 | 53.30 | 41.09 | 42.53 |
| **Dynamic** | | | | | | | | | | | |
| MultiDDS | 47.34 | 41.76 | 21.87 | 63.90 | 66.81 | 42.36 | 44.02 | 47.66 | 52.38 | 39.42 | 36.62 |
| MultiUAT | 47.35 | 36.47 | 23.73 | 65.60 | **67.08** | 43.85 | 47.20 | 49.86 | 52.95 | 44.73 | 41.26 |
| MoS | 47.79 | 44.05 | 19.40 | **65.90** | 66.73 | 42.86 | 48.37 | 53.29 | 53.66 | 44.20 | 42.31 |
| MoS+ | 48.03 | 44.22 | 20.47 | 64.70 | 66.45 | 44.32 | 49.04 | 53.98 | **54.13** | 47.95 | 40.11 |
| *HBO* | **49.37**† | **45.58** | **24.07** | **65.90** | 66.82 | 44.48 | **50.16**† | **55.56** | 53.83 | **48.36** | **42.89** |
| **Llama-3.1-8B** | | | | | | | | | | | |
| **Heuristic** | | | | | | | | | | | |
| Prop. | 46.25 | 40.78 | 18.00 | 62.70 | 64.63 | 45.11 | 46.74 | 51.84 | 51.10 | 43.14 | 40.88 |
| Temp. | 44.78 | 41.97 | 10.33 | 63.50 | 63.95 | 44.16 | 47.68 | 54.34 | 52.93 | 42.91 | 40.55 |
| Uni. | 44.38 | 40.18 | 11.27 | 63.90 | 64.39 | 42.17 | 46.75 | 52.98 | 52.24 | 42.23 | 39.54 |
| **Dynamic** | | | | | | | | | | | |
| MultiDDS | 46.86 | 41.80 | 18.00 | **64.90** | 65.10 | 44.49 | 47.94 | 54.15 | **52.97** | 48.22 | 36.43 |
| MultiUAT | 46.80 | 41.31 | 19.73 | 62.90 | 65.48 | 44.55 | 50.51 | 55.41 | 51.21 | 54.28 | 41.12 |
| MoS | 46.44 | 42.60 | 17.87 | 64.30 | 65.58 | 41.86 | 49.41 | 52.96 | 50.55 | 56.77 | 37.37 |
| MoS+ | 46.94 | 41.25 | 17.73 | 64.10 | 64.96 | 46.64 | 50.94 | 55.55 | 52.69 | 53.07 | 42.43 |
| *HBO* | **48.07**† | **44.28** | **20.40** | 63.00 | **65.98** | **46.67** | **52.28**† | **56.87** | 52.56 | **56.94** | **42.74** |
| **Qwen2.5-7B** | | | | | | | | | | | |
| **Heuristic** | | | | | | | | | | | |
| Prop. | 53.50 | 53.20 | 41.60 | 64.90 | 64.57 | 43.22 | 58.13 | 60.49 | 59.52 | 67.10 | 45.40 |
| Temp. | 54.20 | 51.10 | 46.20 | 65.50 | 64.85 | 43.32 | 54.97 | 62.48 | 49.63 | 62.47 | 45.28 |
| Uni. | 53.90 | 50.46 | 42.93 | 66.30 | 64.92 | **44.90** | 57.42 | 61.52 | 53.66 | 68.31 | 46.19 |
| **Dynamic** | | | | | | | | | | | |
| MultiDDS | 53.82 | 51.84 | 41.80 | **66.70** | 65.42 | 43.36 | 59.27 | 64.81 | 59.16 | 64.90 | **48.22** |
| MultiUAT | 53.79 | 53.13 | 40.40 | 66.40 | 65.10 | 43.91 | 58.90 | 63.66 | 57.33 | 67.70 | 46.93 |
| MoS | 53.99 | 49.80 | 44.87 | 66.10 | **65.72** | 43.48 | 58.84 | 64.79 | 52.75 | 70.28 | 47.55 |
| MoS+ | 54.26 | 51.53 | 44.80 | 66.20 | 65.42 | 43.36 | 58.11 | 63.72 | 53.37 | 70.25 | 45.10 |
| *HBO* | **55.21**† | **53.74** | **48.07** | **66.80** | 64.63 | 42.83 | **60.37**† | **65.25** | **59.93** | **70.35** | 45.94 |

**Model Backbones and Baselines** We evaluate *HBO* on: Qwen2.5-7B (Yang et al., 2024a), Llama-3.1-8B (Dubey et al., 2024), and EuroLLM-9B (Martins et al., 2024). For baselines, we compare against: **Heuristic Methods**: Proportional sampling (*Prop.*, $\tau = 1$), temperature sampling (*Temp.*, $\tau = 10$), and uniform sampling (*Uni.*, $\tau = \infty$), based on Equation 3. **Dynamic Methods**: *MoS* (Wu et al., 2024a), *MultiDDS* (Wang et al., 2020b), and *MultiUAT* (Wu et al., 2021), which adjust dataset distributions globally.

## 4.2 MAIN RESULTS

In this section, we present the performance of *HBO* compared to heuristic and dynamic methods across three LLMs on multilingual and multitask setups, as shown in Table 1. We demonstrate that *HBO* not only significantly outperforms a number of established baselines but also effectively balances the data allocation during the fine-tuning process.

***HBO* consistently outperforms all the baselines in both multilingual and multitask setups.** Results in Table 1 demonstrate that *HBO* consistently outperforms both heuristic and dynamic baselines across various models and evaluation settings. In the multilingual evaluation ($\mu_{\text{ML}}$), *HBO*

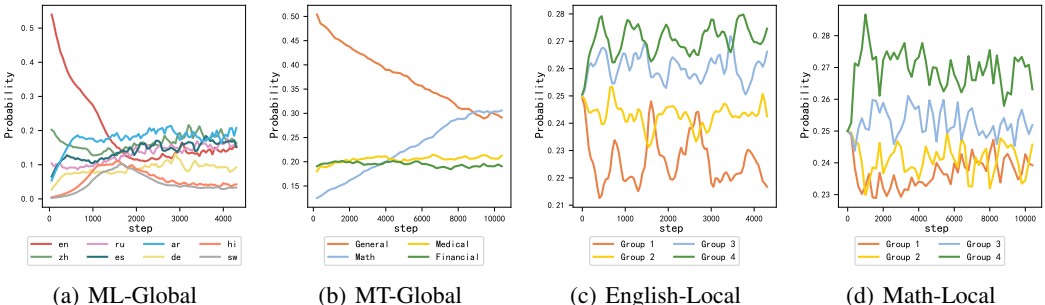

|   (a) ML-Global   |   (b) MT-Global   |   (c) English-Local   |   (d) Math-Local   |

Figure 2: The variation of sampling probabilities given by (a) *global actor* in the multilingual setup, (b) *global actor* in the multitask setup, (c) *local actor* in the English subset, and (d) *local actor* in the Math subset. The model backbone is Llama-3.1-8B.

Table 2: Ablation study of *global actor* and *local actors* given by Llama-3.1-8B in the multilingual setting. ✓ indicates the actor is used, while ✗ indicates the actor is ablated. The values in red and green indicate the gap with the results given by the full *HBO*.

| $\psi_{\text{global}}$ | $\psi_{\text{local}}$ | $\mu_{\text{ML}}$ | MMMLU | MGSM | XCOPA | XStoryCloze | XNLI |
|:---:|:---:|:---:|:---:|:---:|:---:|:---:|:---:|
| ✓ | ✓ | 48.07 | 44.28 | 20.40 | 63.00 | 65.98 | 46.67 |
| ✓ | ✗ | 47.22 ↓0.85 | 43.15 ↓1.13 | 19.47 ↓0.93 | 63.30 ↑0.30 | 65.04 ↓0.94 | 45.14 ↓1.53 |
| ✗ | ✓ | 47.46 ↓0.61 | 43.43 ↓0.85 | 19.13 ↓1.27 | 63.10 ↑0.10 | 65.22 ↓0.76 | 46.44 ↓0.23 |
| ✗ | ✗ | 46.25 ↓1.82 | 40.78 ↓3.50 | 18.00 ↓2.40 | 62.70 ↓0.30 | 64.63 ↓1.35 | 45.11 ↓1.56 |

surpasses the best baselines by margins of +0.87, +1.13, and +0.95 on EuroLLM-9B, Llama-3.1-8B, and Qwen2.5-7B, respectively. Similarly, in the multitask setting ($\mu_{\text{MT}}$), *HBO* delivers superior performance, with gains of +1.06, +1.34, and +1.10 over the best baselines. Beyond average scores, *HBO* achieves even larger performance gains on specific tasks. For example, *HBO* outperforms competing baselines by substantial margins ranging from +1.68 to +4.10 in MMMLU task with the Llama-3.1-8B backbone. These results show *HBO* 's global-local balancing mechanism is crucial for consistently achieving optimal performance across diverse tasks and languages.

***HBO* can effectively balance the data allocation globally and locally.**    We visualize the evolution of sampling probabilities for *global actor* and *local actor* in Figure 2. At the **global** level, *global actor* adaptively balances emphasis between languages or tasks, gradually shifting focus from high-resource (e.g., English or General tasks) to less-represented languages and tasks throughout training (see Figure 2(a) and Figure 2(b)). This dynamic rebalancing prevents overfitting to large datasets while ensuring comprehensive capability development across all languages or tasks. At the **local** level, *local actor* reveals a cyclical pattern in sampling distributions by example difficulty. As illustrated in Figure 2(c), the English subset cycles approximately every 800 steps, where harder examples (groups 3 and 4) and easier examples (groups 1 and 2) alternately receive increased sampling probability. This emergent curriculum learning behavior shows *HBO*'s ability to autonomously adapt for effective training dynamics. The complementary global-local optimization enables *HBO* to simultaneously address dataset imbalance and heterogeneity at multiple levels, resulting in superior performance.

## 5 ANALYSIS

In this section, we evaluate *HBO* in a multilingual setting, examining actor ablation, different difficulty groupings, various sampling priors, and training data sizes. Further analyses on model sizes, reward functions and updating frequency are provided in Appendix B.

**Both *global actor* and *local actors* can effectively improve performance.**    We conduct an ablation study to assess the individual contributions of *global actor* and *local actors*. The results in Table 2 demonstrate that ablating *local actors* reduces $\mu_{\text{ML}}$ by 0.85 and XNLI by 1.53, while removing *global*

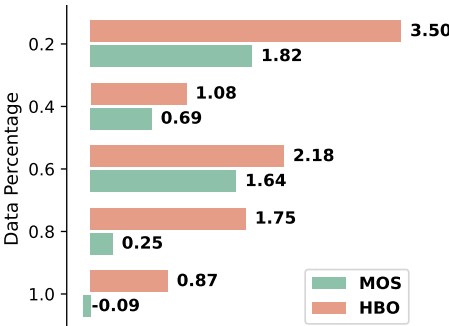

Figure 3: Improvements of *HBO* and *MoS* compared to the *Prop.* with Llama-3.1-8B backbone on the MMMLU testset.

Table 4: Effect of different sampling priors ($\tau$) on performance using Llama-3.1-8B backbone across multilingual benchmarks. The best results are highlighted in **bold**.

|  | $\mu_{\mathrm{ML}}$ | MMMLU | MGSM | XCOPA | XSC | XNLI |
|---|---|---|---|---|---|---|
| *Prop.* | 46.25 | 40.78 | 18.00 | 62.70 | 64.63 | 45.11 |
| *MoS* | | | | | | |
| $+\tau = 1$ | 46.44 | 42.60 | 17.87 | 64.30 | 65.58 | 41.86 |
| $+\tau = 10$ | 47.06 | 42.80 | 18.00 | **64.90** | 65.10 | 44.49 |
| $+\tau = \infty$ | 46.40 | 41.31 | 17.73 | 62.90 | 65.48 | 44.55 |
| *HBO* | | | | | | |
| $+\tau = 1$ | **48.07** | 44.28 | 20.40 | 63.00 | 65.98 | **46.67** |
| $+\tau = 10$ | 47.98 | 44.30 | **20.80** | 63.60 | 65.92 | 45.30 |
| $+\tau = \infty$ | 48.06 | **44.64** | 19.73 | 63.30 | **66.05** | 46.59 |

*actor* lowers $\mu_{\mathrm{ML}}$ by 0.61 and MGSM by 1.27. Removing both actors causes the largest drop (1.82 on $\mu_{\mathrm{ML}}$ and 3.50 on MMMLU), confirming that each type of actor offers distinct advantages and that their combination optimally balances global and local distributions during training.

**The optimal number of difficulty levels balances granularity and effective learning.** We examine the impact of difficulty level granularity on *HBO* by partitioning each dataset into 1, 2, 4, 8, and 16 groups. As shown in Table 3, four-level grouping consistently yields the best overall performance, achieving an $\mu_{\mathrm{ML}}$ score of 48.07. The pattern suggests that moderate granularity provides an optimal balance, as too few levels fail to capture meaningful distinctions between examples, while excessive divisions may fragment the learning signal and increase the complexity.

Table 3: Difficulty level granularity with Llama-3.1-8B. Best results are in **bold**.

| #Groups | $\mu_{\mathrm{ML}}$ | MMMLU | MGSM | XCOPA | XSC | XNLI |
|---|---|---|---|---|---|---|
| 1 | 47.22 | 43.15 | 19.47 | 63.30 | 65.04 | 45.14 |
| 2 | 47.09 | 44.09 | 18.07 | 63.60 | 65.52 | 44.16 |
| 4 | **48.07** | **44.28** | **20.40** | 63.00 | **65.98** | **46.67** |
| 8 | 47.45 | 44.16 | 18.53 | 63.80 | 65.06 | 45.68 |
| 16 | 47.87 | 43.92 | 20.33 | **64.00** | 65.60 | 45.48 |

***HBO* shows robust improvements across sampling priors.** Dynamic methods are often sensitive to the choice of prior sampling distribution (Wu et al., 2021). We evaluate *HBO* at various temperature settings in Table 4 and find that *HBO* consistently outperforms the best heuristic baseline *Prop.* across all tasks, with gains of up to +1.82 in $\mu_{\mathrm{ML}}$. While the optimal prior may vary by task, *HBO* remains stable and shows significantly less variance than *MoS*, demonstrating its robustness against changes in the prior sampling distribution and consistent performance gains across diverse evaluation scenarios.

***HBO* consistently outperforms the baselines even in resource-constrained environments.** We previously use only 20% of the available training data to test our method under constrained conditions, and now we increase data usage to 40%, 60%, 80%, and 100%. As shown in Figure 3, *HBO* consistently achieves larger performance gains compared to *MoS* in every setting. Notably, when using only 20% of the training data, *HBO* delivers a performance boost of 3.50 over *Prop.*.

**Easy examples matter for model performance.** To investigate the importance of easy examples in training, we progressively discard increasing percentages of the easiest training examples while maintaining the total training compute (e.g., doubling the number of training steps when 50% of examples were discarded). Our findings in Table 5 show consistent performance degradation as more easy examples are discarded. Removing 25% of the easiest examples causes a noticeable drop

Table 5: Performance comparison when progressively discarding the easiest examples from the training dataset with Llama-3.1-8B. The "pct." is the percentage of easiest examples discarded from training set.

|  | pct. | $\mu_{\mathrm{ML}}$ | MMMLU | MGSM | XCOPA | XSC | XNLI |
|---|---|---|---|---|---|---|---|
| *Prop.* | 0% | 46.25 | 40.78 | 18.00 | 62.70 | 64.63 | 45.11 |
| *HBO* | 0% | 48.07 | 44.28 | 20.40 | 63.00 | 65.98 | 46.67 |
| | 25% | 47.41 | 43.08 | 21.00 | 62.30 | 65.94 | 44.75 |
| | 50% | 47.08 | 41.32 | 22.33 | 62.60 | 65.72 | 43.45 |
| | 75% | 46.55 | 41.09 | 18.80 | 66.10 | 63.20 | 43.58 |

in average performance (-0.66 on $\mu_{\text{ML}}$), particularly on XNLI (-1.92). As removal reaches 75%, performance nears that of the baseline. Although easy examples are often considered less informative (Xu et al., 2024), they prove crucial by diversifying the training mixture.

## 6 Conclusion

In this paper, we present ***Hierarchical Balancing Optimization (HBO)***, a novel hierarchical dynamic data sampling method designed to tackle the critical challenges of data imbalance and heterogeneity in fine-tuning LLMs. Leveraging a bilevel optimization framework with a **Global Actor** and several **Local Actors**, *HBO* enables LLMs to autonomously adjust data sampling both across datasets (globally) and within datasets (locally) based on their current training state. Through extensive experiments across three LLMs and nine tasks in multilingual and multitask setups, we demonstrate the effectiveness of *HBO*, achieving significant performance improvements. By autonomously adapting LLMs' learning strategies, *HBO* represents a significant advancement in addressing the complexities of dataset mixture balancing, contributing to more effective fine-tuning of LLMs.

## Ethics Statement

This work introduces *HBO*, a method for fine-tuning large language models (LLMs) using hierarchical dynamic data sampling. All experiments are conducted on publicly available datasets and open-source model backbones, strictly adhering to their respective licenses and terms of use. No human subjects or private data are involved. While *HBO* aims to improve fairness and generalization by addressing data imbalance and heterogeneity, we acknowledge that biases present in the underlying datasets or models may persist. We encourage responsible use of *HBO*, with attention to fairness, transparency, and accountability in downstream applications.

## Reproducibility Statement

We are committed to ensuring the reproducibility of our findings. Detailed descriptions of the *HBO* methodology, including the bilevel optimization framework, actor architectures, reward functions, and training procedures, are provided in Section 3. Experimental setups, including model backbones, dataset statistics, sampling strategies, and evaluation metrics, are thoroughly described in Section 4 and Appendix A. All datasets and models used are publicly available, with references and links included. To further support reproducibility, we will release our code and scripts for data preparation and experiments upon publication, enabling other researchers to replicate and build upon our results.

## The Use of Large Language Models (LLMs)

In preparing this work, we utilize large language models (LLMs) as general-purpose tools to assist with writing polish and grammar correction. The LLMs are not involved in research ideation, experimental design, or substantive content generation. Their role is limited to improving the clarity and readability of the text, ensuring grammatical accuracy, and refining the presentation of our findings. All scientific contributions, analyses, and conclusions are solely the work of the authors.

## Acknowledgement

This project has received funding from UK Research and Innovation (UKRI) under the UK government's Horizon Europe funding guarantee [grant numbers 10039436].

The computations described in this research were performed using the Baskerville Tier 2 HPC service (https://www.baskerville.ac.uk/). Baskerville was funded by the EPSRC and UKRI through the World Class Labs scheme (EP/T022221/1) and the Digital Research Infrastructure programme (EP/W032244/1) and is operated by Advanced Research Computing at the University of Birmingham.

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

## A  TRAINING DETAILS

We fine-tune all parameters of LLMs using the AdamW optimizer with a learning rate of $1 \times 10^{-5}$ and a batch size of 16. This process is conducted over three epochs on 8 NVIDIA A100 GPUs (80GB). During training, we use a linear learning rate schedule with a warm-up phase that constitutes 10% of the total training steps. For *HBO*, $\psi_{\text{global}}$ and $\psi_{\text{local}}$ are updated for every 200 steps with the learning rate of $1 \times 10^{-4}$ and the batch size of 64. $\psi_{\text{global}}$ and $\psi_{\text{local}}$ are initialized by $\tau = 1$.

## B  ADDITIONAL ANALYSIS

**Performance gains scale with parameter count.** As shown in Table 6, we observe that *HBO* consistently outperforms heuristic methods across all model sizes. Notably, the performance advantage scales with the model's parameter count, suggesting that larger models are better able to leverage the optimized sampling strategies. For average performance ($\mu_{\text{ML}}$), *HBO* demonstrates a significant improvement over *Prop.*, achieving a +1.83 gain with Llama-3.1-8B, compared to smaller gains of +1.25 with the 3B model and +0.40 with the 1B model. While the extent of these improvements may vary depending on the specific tasks, the consistent advantage demonstrated by *HBO* underscores the robustness of this approach. These results indicate that our balancing method is effective across different model sizes, with larger models benefiting more sig-

Table 6: Comparisons between *HBO* and heuristic methods with various model sizes in the multilingual setup.

|  | $\mu_{\text{ML}}$ | MMMLU | MGSM | XCOPA | XSC | XNLI |
|---|---|---|---|---|---|---|
| **Llama-3.2-1B** | | | | | | |
| *Prop.* | 37.60 | 27.43 | 3.53 | 58.30 | 58.00 | 40.73 |
| *Temp.* | 37.59 | 27.60 | **3.67** | 58.40 | 57.58 | 40.70 |
| *Uni.* | 37.62 | 27.68 | 3.20 | **58.60** | 57.51 | 41.13 |
| *HBO* | **38.00** | **28.27** | 3.40 | **58.60** | **58.22** | **41.53** |
| **Llama-3.2-3B** | | | | | | |
| *Prop.* | 41.90 | 37.26 | 7.47 | 60.40 | 61.45 | 42.91 |
| *Temp.* | 42.21 | 37.21 | 7.67 | **61.50** | **62.13** | 42.55 |
| *Uni.* | 42.32 | **38.26** | 8.20 | 60.80 | 61.79 | 42.57 |
| *HBO* | **43.15** | 38.23 | **10.93** | 61.30 | 62.08 | **43.18** |
| **Llama-3.1-8B** | | | | | | |
| *Prop.* | 46.24 | 40.78 | 18.00 | 62.70 | 64.63 | 45.11 |
| *Temp.* | 44.78 | 41.97 | 10.33 | 63.50 | 63.95 | 44.16 |
| *Uni.* | 44.38 | 40.18 | 11.27 | **63.90** | 64.39 | 42.17 |
| *HBO* | **48.07** | **44.28** | **20.40** | 63.00 | **65.98** | **46.67** |

nificantly. This is likely because larger models often possess a greater capacity to leverage advanced optimization techniques.

**HBO is compatible with various reward functions.** The choice of reward function significantly affect the model performance, so we investigate the impact of reward functions and present the results in Table 7. The $L_2$ norm and *PPL Ratio* are the default global and local reward function as defined in Equation 7 and Equation 8, respectively. Following Wu et al. (2024a), we introduce *CosSim* as additional global reward and *PPL* and *Loss* as additional local rewards. The *CosSim* is defined as the cosine similar-

Table 7: Various combinations of reward functions in *HBO* using Llama-3.1-8B under multilinguao setup.

| $\mathcal{R}_{\text{global}}$ | $\mathcal{R}_{\text{local}}$ | $\mu_{\text{ML}}$ | MMMLU | MGSM | XCOPA | XSC | XNLI |
|---|---|---|---|---|---|---|---|
| $L_2$ norm | PPL Ratio | 48.07 | 44.28 | 20.40 | 63.00 | 65.98 | 46.67 |
| $L_2$ norm | PPL | 46.94 | 43.18 | 17.81 | 63.30 | 65.08 | 45.31 |
| $L_2$ norm | Loss | 47.22 | 44.11 | 18.17 | 62.70 | 64.77 | 46.34 |
| CosSim | PPL Ratio | 47.69 | 43.36 | 19.33 | 64.00 | 65.33 | 46.44 |
| CosSim | PPL | 47.05 | 42.89 | 18.50 | 62.80 | 65.08 | 45.97 |
| CosSim | Loss | 47.15 | 43.45 | 18.27 | 63.10 | 64.63 | 46.32 |

ity between the hidden states from two batches, the *PPL* and *Loss* are defined in Equation 8 and Equation 1, respectively. We observe that all the combinations of reward function achieve performance gains compared to the best heuristic baseline *Prop.* (46.24) in $\mu_{\text{ML}}$, and the combination of $L_2$ norm and *PPL Ratio* achieves the best performance among all these combination. These findings validate the effectiveness of our design choice.

**The updating frequency of *global actor* and *local actor* should be carefully determined.** We conduct an experiment to investigate the impact of the updating frequency on the runtime and model performance, and present the results in Figure 4. As shown in Figure 4(a), we observe that *global actor* and *local actor* consistently improve the model performance across all the updating frequency

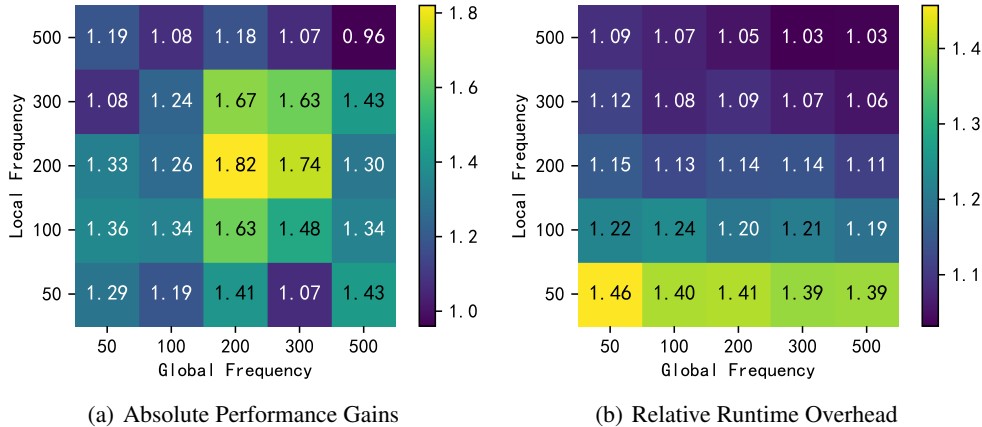

(a) Absolute Performance Gains     (b) Relative Runtime Overhead

Figure 4: (a) The absolute performance gains of *HBO* compared to *Prop.* with different settings of updating frequency for *global actor* and *local actor*. (b) The relative runtime overhead introduced by *HBO* compared to *Prop.* with different settings of updating frequency for *global actor* and *local actor*.

settings and achieve the best performance when setting the updating frequency of both *global actor* and *local actor* to 200. Furthermore, more frequent updating results in more computational overhead. Figure 4(b) demonstrates that a frequency of 200 for both *global actor* and *local actor* provides the best balance between performance gains and computational efficiency.

## C  ADDITIONAL BASELINES COMPARISON

There are several additional baselines that we compare against to further validate the effectiveness of *HBO*. As the code of these works is not open-sourced, we re-implement the baselines DMT (Dong et al., 2024) and EVIC (Liang et al., 2025) and conduct additional experiments using Llama-3.1-8b backbone in our multilingual setting. As shown in Table 8, we observe that HBO outperforms both DMT and EVIC across all the benchmarks.

Table 8: Comparison with additional baselines using Llama-3.1-8b backbone in multilingual setting.

| Methods | $\mu_{\mathrm{ML}}$ | MMMLU | MGSM | XCOPA | XStoryCloze | XNLI |
|---------|------|-------|------|-------|-------------|------|
| DMT | 45.43 | 40.26 | 17.73 | 62.10 | 64.85 | 42.23 |
| EVIC | 46.10 | 41.76 | 19.73 | 62.70 | 63.95 | 42.36 |
| HBO | 48.07 | 44.28 | 20.40 | 63.00 | 65.98 | 46.67 |

## D  METRICS TO DIVIDE DATASETS

In Section 3.2, we use SuperFiltering metric to evenly divide each subset into four groups based on task difficulty. This metric essentially is Instruction Following Difficulty (IFD) score (Li et al., 2024b) but computed using a smaller language model, which is Qwen2.5-0.5B in this work. SuperFiltering demonstrates that a small language model is equally effective in computing IFD scores for data selection compared to using a large language model. IFD score is a pure statistical metric, that compares the losses or perplexities when the model generates a response $y_i$ with and without instructional context $x_i$, measuring how much help the instruction provides to the generation of the corresponding response. For a given instruction-following data pair, the IFD score is calculated as: $\mathrm{IFD}(y_i|x_i) = \frac{\mathrm{PPL}(y_i|x_i)}{\mathrm{PPL}(y_i)}$, where $\mathrm{PPL}(y_i|x_i)$ and $\mathrm{PPL}(y_i)$ denote the perplexities of the given model in fitting response $y_i$ with and without the instruction $x_i$, respectively. A higher IFD score, indicating less instructional help, suggests a greater difficulty. On the contrary, the low IFD score represents that

the given instruction can directly benefit the language model largely even without further training, representing the easiness and necessity of the instruction. For instance, if $x_i$ contains strong clues or direct answers, the model can easily generate $y_i$ with $x_i$, leading to a low IFD score.

Regarding robustness, we conduct additional experiments with using PPL and loss as metrics. The results with Llama-3.1-8b backbone in the multilingual setup are shown in Table 9. We observe that both of HBO-PPL and HBO-Loss achieve the performance gains compared to the best heuristic baseline Prop. It demonstrates that as long as the metric provides a meaningful signal of difficulty, the local actors will learn to balance them effectively.

Table 9: Comparison with different metrics for dataset division.

| Metrics | $\mu_{\mathrm{ML}}$ | MMMLU | MGSM | XCOPA | XStoryCloze | XNLI |
|---|---|---|---|---|---|---|
| Prop. | 46.25 | 40.78 | 18.00 | 62.70 | 64.63 | 45.11 |
| HBO-SuperFiltering | 48.07 | 44.28 | 20.40 | 63.00 | 65.98 | 46.67 |
| HBO-PPL | 47.42 | 44.71 | 19.33 | 62.60 | 64.65 | 45.81 |
| HBO-Loss | 47.23 | 44.88 | 19.04 | 62.50 | 64.39 | 45.34 |

