# OpenReview forum: "HBO: Hierarchical Balancing Optimization for Fine-Tuning Large Language Models"
_ICLR.cc/2026/Conference — ICLR 2026 Poster_

### Official Review · Reviewer_9Boj · 2025-10-28

**Soundness:** 3
**Presentation:** 3
**Contribution:** 2
**Rating:** 6
**Confidence:** 3

**Summary:**

This paper introduces Hierarchical Balancing Optimization (HBO), a novel framework designed to address data imbalance and heterogeneity challenges in fine-tuning LLMs. HBO employs a bilevel optimization strategy with two key components: a Global Actor for balancing data sampling across datasets and Local Actors for optimizing sampling within individual datasets based on difficulty levels. The paper demonstrates HBO’s effectiveness across three LLM architectures on multilingual and multitask setups, showing consistent performance improvements over existing baselines.

**Strengths:**

The hierarchical optimization framework with global and local balancing is novel and addresses a critical gap in LLM fine-tuning.
The experiments are extensive and well-designed, covering multiple models, tasks, and evaluation metrics. The ablation studies and sensitivity analyses provide deep insights into the effectiveness of HBO.
The paper is clearly written, with detailed explanations of the methodology and results. Figures and tables are well-designed and enhance understanding.
The proposed method is broadly applicable and demonstrates consistent performance gains, making it a valuable contribution to the field.

**Weaknesses:**

While HBO demonstrates strong performance improvements, the additional 15% training overhead may limit its applicability in resource-constrained environments. The authors could provide more discussion on strategies to mitigate this overhead.

Although the paper evaluates different reward functions, the focus is primarily on L2 norm and perplexity ratio. Exploring alternative reward mechanisms (e.g., task-specific metrics) could further enhance HBO’s applicability.

The experiments are conducted on sampled datasets (e.g., 20% of total data). It would be helpful to discuss how HBO scales to larger datasets and whether the performance gains remain consistent.

**Questions:**

Have the authors considered task-specific reward functions or alternative optimization techniques for the global and local actors? If so, how do these compare to the current design?

---

> ### Author Response · Authors · 2025-11-19
> **Response to Reviewer 9Boj**
>
> Thank you for your detailed review of our paper. We appreciate your insights and the opportunity to clarify and strengthen our work. Below, we address each of your concerns individually.
>
>
> > **Q1:** While HBO demonstrates strong performance improvements, the additional 15\% training overhead may limit its applicability in resource-constrained environments. The authors could provide more discussion on strategies to mitigate this overhead.
> > >
> > > **R1:** We thank the reviewer for this practical concern. We were transparent about the ~15\% overhead, which we consider a modest trade-off for the significant accuracy gains.
> > >
> > > However, this overhead is not fixed; it is a tunable parameter. The 15\% cost corresponds to our chosen update frequency of 200 steps for both actors. We explicitly analyze the trade-off between performance and runtime overhead in Appendix B, Figure 4.
> > >
> > > As shown in Figure 4(b), the runtime overhead decreases as the update frequency becomes less frequent. A user in a resource-constrained environment could easily mitigate this overhead by choosing a less frequent update schedule (e.g., every 500 steps), which still provides performance gains (as shown in Figure 4(a)) but with a much lower computational cost. Our default setting of 200 steps was chosen because it offered the "best balance between performance gains and computational efficiency".
>
>
>
> > **Q2:** Although the paper evaluates different reward functions, the focus is primarily on L2 norm and perplexity ratio. Exploring alternative reward mechanisms (e.g., task-specific metrics) could further enhance HBO’s applicability.
> > >
> > > **R2:** We agree that exploring different reward functions is critical. In Appendix B, We get the conclusion: "HBO is compatible with various reward functions" and Table 7, we conduct an extensive ablation study precisely on this point. We tested CosSim as an alternative global reward. We tested both PPL and Loss as alternative local rewards.
> > >
> > > The results in Table 7 show that all tested combinations of reward functions achieved performance gains compared to the heuristic baseline (Prop.). While our chosen combination of $L_2$ norm and PPL Ratio achieved the best performance, this analysis demonstrates the robustness of the HBO framework itself, which is not dependent on a single reward design. As the field progresses, new reward functions may emerge, and HBO can easily incorporate these advancements.
> > >
> > > For the use of task-specific reward functions, please refer to the discussion in R4.
>
>
> > **Q3:** The experiments are conducted on sampled datasets (e.g., 20\% of total data). It would be helpful to discuss how HBO scales to larger datasets and whether the performance gains remain consistent.
> > >
> > > **R3:** The main experiments used 20\% data samples due to computational constraints. To address the question of scaling, We have conducted a dedicated analysis in Section 5, shown in Figure 3. In this experiment, we varied the data usage from 20\% up to 100\% (the full dataset). The results clearly show that "HBO consistently achieves larger performance gains compared to MoS in every setting". This analysis confirms that our performance gains remain consistent and, in fact, are highly pronounced at different data scales, including the full dataset.
>
> > **Q4:** Have the authors considered task-specific reward functions or alternative optimization techniques for the global and local actors? If so, how do these compare to the current design?
> > >
> > > **R4:** Thank you for your valuable comments. We did not consider task-specific reward functions in this work, as our focus was on designing a general framework applicable across multiple datasets and tasks. However, we are happy to discuss the potential of task-specific rewards in this rebuttal.
> > >
> > > Using task-specific reward functions for local actors could potentially enhance HBO's performance, but not for the global actor. The global actor is responsible for balancing across datasets/tasks. In the implementation, we could certainly use task-specific metrics (e.g., accuracy for classification tasks, BLEU for generation tasks) as global rewards. However, these rewards are not directly comparable across different tasks, making it challenging for the global actor to learn a coherent policy. In contrast, task-specific reward functions are likely to be more suitable for local actors, as the rewards across different difficulty groups within the same dataset/task are comparable. Our current design present a more generalized approach that works well across diverse datasets and tasks.
> > > We will explore this direction in future work and include a discussion of this potential extension in our revision.

---

### Official Review · Reviewer_9Qbd · 2025-10-29

**Soundness:** 2
**Presentation:** 3
**Contribution:** 2
**Rating:** 4
**Confidence:** 4

**Summary:**

This paper presents Hierarchical Balancing Optimization (HBO), a bilevel optimization framework designed to mitigate data imbalance and heterogeneity when fine-tuning LLMs. HBO introduces two interacting components, i.e., a Global Actor that adjusts sampling probabilities across datasets and several Local Actors that manage sampling within datasets based on difficulty levels. Both actors are guided by rewards derived from the LLM's training state: gradient norms for global balancing and perplexity ratios for local balancing. The authors evaluate HBO on three model backbones (Llama-3.1-8B, Qwen2.5-7B, EuroLLM-9B) and nine tasks across multilingual and multitask setups, reporting consistent performance improvements over both heuristic and dynamic baselines.

**Strengths:**

1. The paper targets an important problem, i.e., data imbalance and heterogeneity in LLM fine-tuning,which is relevant to current multi-task and multilingual training paradigms.
2. The hierarchical bilevel optimization formulation is conceptually interesting and provides a unified framework for global and local data balancing.
3. The paper is well-written and easy to follow.

**Weaknesses:**

I have the following concerns. *If the authors could properly address them during the rebuttal phase, I am willing to raise my score.*
1. The technical novelty is somewhat limited. While the hierarchical structure and bilevel setup are well-motivated, they mainly combine known techniques such as policy gradients and dynamic sampling into a straightforward framework, without introducing fundamentally new optimization principles.
2. This paper lacks strong theoretical or analytical justification for why the proposed reward formulations (gradient norm and perplexity ratio) should lead to optimal or stable data allocation.
3. Although the experiments are comprehensive, the reported improvements are modest (often within 1–2 points), raising questions about whether the additional implementation overhead (15% extra training cost) is justified.
4. Some important recent baselines [1,2] are missing. The paper would benefit from either a comparison with these methods or a discussion of their relevance.
5. The authors may want to provide deeper analysis of failure cases or scenarios where HBO underperforms, as well as qualitative insights into how the hierarchical balancing actually affects learning dynamics beyond probability curves.

[1] How Abilities in Large Language Models are Affected by Supervised Fine-tuning Data Composition. ACL 2024.

[2] Boosting Multi-Domain Fine-Tuning of Large Language Models through Evolving Interactions between Samples. ICML 2025.

**Questions:**

Please see Weaknesses.

---

> ### Author Response · Authors · 2025-11-19
> **Response to Reviewer 9Qbd (part1)**
>
> Thank you for your thorough review of our paper and for the insightful questions you've raised. We appreciate the opportunity to address your concerns and clarify various aspects of our work. Below, we provide detailed responses to each of your points.
>
>
> > **Q1:** The technical novelty is somewhat limited. While the hierarchical structure and bilevel setup are well-motivated, they mainly combine known techniques such as policy gradients and dynamic sampling into a straightforward framework, without introducing fundamentally new optimization principles.
> > >
> > > **R1:** We respectfully disagree with the characterization of the paper's novelty as limited. While we agree that components like policy gradients are well-established, our core technical contribution is not the invention of these components, but their novel application within a hierarchical bilevel framework to solve a previously unaddressed problem in LLM fine-tuning.
> > >
> > > The novelty lies in:
> > > - Identifying the Problem: Prior work has almost exclusively focused on "global" balancing (across datasets). We are the first to identify and systematically tackle the critical issue of "local" imbalance and heterogeneity (within individual datasets).
> > > - A Novel Framework: Our hierarchical structure, which employs a Global Actor for inter-dataset balancing and distinct Local Actors for intra-dataset balancing, is a total new method designed specifically for this dual-level problem. This bilevel optimization is not a "straightforward" combination; it is a motivated architecture to manage the distinct reward signals and optimization goals at each level of the data hierarchy.
> > > - Comprehensive Evaluations: We provide extensive empirical validation across multiple models and tasks, demonstrating that this hierarchical approach yields significant performance gains over both static and prior dynamic methods.
> > > Combining these factors together, we believe our work has made substantial and novel contributions to LLM fine-tuning.
>
>
> > **Q2:** This paper lacks strong theoretical or analytical justification for why the proposed reward formulations (gradient norm and perplexity ratio) should lead to optimal or stable data allocation.
> > >
> > > **R2:** We thank the reviewer for this point, as it allows us to clarify the strong motivation behind our reward functions. While we did not provide a formal proof of "optimality", our reward functions are not arbitrary. They are well-justified by prior work and empirically validated by our own ablations.
> > >
> > > Global Reward (Gradient Norm): This choice is based on established findings that the $L_2$ norm of gradients serves as an "ideal signal of learning dynamics", as evidenced by [1]. A larger gradient norm indicates that the model has more to learn from a specific subset, making it a principled signal for allocating global resources.
> > >
> > > Local Reward (Perplexity Ratio): This reward is designed to measure relative learning progress, following [2]. By using a ratio of current to initial perplexity, we "prioritize examples based on their learning progress". This prevents the model from getting stuck on intractably hard examples while also not oversampling examples it has already mastered.
> > >
> > > Furthermore, we empirically validated these choices in Appendix B, Table 7. This ablation shows that while other reward combinations also provide gains, our chosen pairing of $L_2$ norm and PPL Ratio achieves the best performance, validating our design choices.
> > >
> > > References:
> > > 1. Chen, Zhao, Vijay Badrinarayanan, Chen-Yu Lee, and Andrew Rabinovich. "Gradnorm: Gradient normalization for adaptive loss balancing in deep multitask networks." In International conference on machine learning, pp. 794-803. PMLR, 2018.
> > > 2. Wu, Minghao, Thuy Vu, Lizhen Qu, and Reza Haf. "Mixture-of-skills: Learning to optimize data usage for fine-tuning large language models." In Proceedings of the 2024 Conference on Empirical Methods in Natural Language Processing, pp. 14226-14240. 2024.

---

> ### Author Response · Authors · 2025-11-19
> **Response to Reviewer 9Qbd (part2)**
>
> > **Q3:** Although the experiments are comprehensive, the reported improvements are modest (often within 1–2 points), raising questions about whether the additional implementation overhead (15\% extra training cost) is justified.
> > >
> > > **R3:** We respectfully argue that the performance gains are significant in the context of LLM fine-tuning, and the 15\% overhead is a modest and acceptable trade-off.
> > >
> > > While some average gains are 1-2 points, these are macro-averages over strong heuristic and dynamic baselines. In this competitive domain, consistent gains of +0.87 to +1.34 $\mu_{ML}$/$\mu_{MT}$ over the best baseline across three different models are substantial. More importantly, on specific, challenging tasks, the gains are much larger. For example, HBO outperforms all baselines by +1.68 to +4.10 points on the MMMLU task with the Llama-3.1-8B model. We conducted bootstrap-based statistical significance tests [1] across multiple runs. The results confirm that the improvements achieved by HBO over the best baselines are statistically significant (p < 0.05) in all three model backbones.
> > >
> > > A 15\% increase in training runtime 14 is a very reasonable price for these **consistent and significant** accuracy improvements. Moreover, as shown in Appendix B, Figure 4, this 15\% overhead is not fixed. It is a result of our chosen update frequency (200 steps), which we found to be the "best balance between performance gains and computational efficiency". A user who is more resource-constrained could choose a less frequent update schedule to reduce this overhead further.
> > >
> > > References:
> > > 1. Philipp Koehn. 2004. Statistical Significance Tests for Machine Translation Evaluation. In Proceedings of the 2004 Conference on Empirical Methods in Natural Language Processing, pages 388–395, Barcelona, Spain. Association for Computational Linguistics.
>
>
> > **Q4:** Some important recent baselines [1,2] are missing. The paper would benefit from either a comparison with these methods or a discussion of their relevance.
> >
> > [1] How Abilities in Large Language Models are Affected by Supervised Fine-tuning Data Composition. ACL 2024.
> >
> > [2] Boosting Multi-Domain Fine-Tuning of Large Language Models through Evolving Interactions between Samples. ICML 2025.
> > >
> > > **R4:** We thank the reviewer for bringing these recent papers to our attention. As the code of these works is not open-sourced, we re-implement the baselines DMT[1] and EVIC[2] and conduct additional experiments using Llama-3.1-8b backbone in our multilingual setting. As shown in the following table, we observe that HBO outperforms both DMT and EVIC across all the benchmarks.
> > >
> |   Metric    | $\mu_{ML}$ |   MMMLU   |    MGSM   |   XCOPA   |    XStoryCloze    |     XNLI     |
> |---|---|---|---|---|---|---|
> |         DMT     |    45.43    |     40.26  |   17.73   |     62.10 |     64.85 |     42.23   |
> |         EVIC     |    46.10    |     41.76  |   19.73   |     62.70 |     63.95 |     42.36   |
> |         HBO   |      48.07  |     44.28 |     20.40 |     63.00 |     65.98 |     46.67    |

---

> ### Author Response · Authors · 2025-11-19
> **Response to Reviewer 9Qbd (part3)**
>
> > **Q5:** The authors may want to provide deeper analysis of failure cases or scenarios where HBO underperforms, as well as qualitative insights into how the hierarchical balancing actually affects learning dynamics beyond probability curves.
> > >
> > > **R5:** We appreciate the suggestion for deeper analysis. We believe our paper already provides significant qualitative insights into the learning dynamics beyond simple probability curves, particularly in Section 5.
> > >
> > > We conducted a direct analysis of how balancing affects learning by progressively discarding the easiest 25\%, 50\%, and 75\% of data in Table 5. The "consistent performance degradation" we observed provides a crucial insight: HBO's ability to "diversify the training mixture" by not discarding easy examples is critical to its success. This is a direct qualitative analysis of why our balancing works.
> > >
> > > As shown in Figure 2, the discussion of the cyclical pattern in the local actors is an analysis of the learning dynamics. We show that the model learns an "emergent curriculum", alternating focus between easy and hard groups. This demonstrates how the hierarchical balancing autonomously adapts its strategy during training.
> > >
> > > As for failure cases, we observe that dynamic rebalancing is not necessary when the heterogeneity and imbalance are minimal in our preliminary study. For instance, when the subsets in the dataset mixture are well-balanced, the gains from MultiDDS, MultiUAT, MoS, and HBO are minimal compared to the heuristic baselines. This is expected, as there is no need to rebalance when the data is already balanced. This suggests that the benefits of HBO are most pronounced in scenarios with significant imbalance and heterogeneity, which is common in real-world multi-dataset fine-tuning scenarios. We will include this discussion in our revision to provide a more balanced view of the method's applicability.

---

> ### Comment · Reviewer_9Qbd · 2025-11-20
> **Thank you for the detailed rebuttal.**
>
> Thank you for the detailed rebuttal, which has largely addressed my concerns.
>
> However, one important issue remains: **the authors have not yet updated the paper.** I hope the authors will make full use of ICLR's allowance to update the submission during the rebuttal phase. I look forward to seeing an improved version of the paper, **incorporating the discussions and explanations from the rebuttal as well as the additional experiments and baselines**. At that point, I will consider increasing my score.

---

> > ### Author Response · Authors · 2025-11-21
> > **Response to Reviewer 9Qbd**
> >
> > Thank you for your further comments. We have updated our paper to include the additional experiments and discussions. The updated content is highlighted in the orange text. To preserve the original structure for discussion with other reviewers, we update the caption of table 1 to include statistical significance results and add the new experiments and discussions in Appendix C, D, and E. We will relocate these results to the main paper in the final version to highlight their importance.

---

> > > ### Comment · Reviewer_9Qbd · 2025-11-21
> > > **My concerns have been addressed.**
> > >
> > > Thank you for your response. After reading the updated paper, I believe that my concerns have been addressed. Therefore, I have raised my score to 6 and am willing to support the acceptance of this paper.

---

### Official Review · Reviewer_9MNG · 2025-10-29

**Soundness:** 3
**Presentation:** 3
**Contribution:** 3
**Rating:** 6
**Confidence:** 4

**Summary:**

This paper introduces Hierarchical Balancing Optimization (HBO), a novel framework designed to address data imbalance and heterogeneity during the fine-tuning of Large Language Models (LLMs). The authors argue that existing methods typically focus on balancing data across different datasets (globally) but neglect imbalances and variations within individual datasets (locally). HBO aims to solve this by enabling the LLM to autonomously adjust data sampling probabilities at both levels. It employs a bilevel optimization strategy featuring a Global Actor, responsible for balancing sampling across datasets, and multiple Local Actors, each optimizing sampling within a specific dataset based on predefined difficulty groups. These actors are trained using reward signals derived from the LLM's training state: the global reward is based on the L2 norm of gradients (indicating learning progress), and the local reward uses the ratio of current to initial perplexity (indicating relative improvement on difficulty groups). The framework is evaluated on several LLM backbones across multilingual and multitask settings, reportedly outperforming various baseline sampling strategies.

**Strengths:**

1. The paper tackles a significant and nuanced challenge in LLM fine-tuning by explicitly addressing hierarchical data imbalance and heterogeneity (both global, across datasets, and local, within datasets), which is often overlooked by simpler methods.

2. The proposed HBO mechanism, utilizing a bilevel optimization framework with distinct global and local actors guided by rewards derived from the model's own training state, is a novel and sophisticated approach to achieve autonomous, dynamic data balancing.

3. The empirical results are strong and comprehensive, showing consistent improvements over heuristic and existing dynamic sampling baselines across multiple LLM backbones (Llama, Qwen, EuroLLM), diverse tasks (multilingual, multitask), and including insightful analyses like the dynamic evolution of sampling probabilities.

**Weaknesses:**

1. The framework introduces substantial complexity compared to standard fine-tuning or simpler dynamic sampling. Implementing and tuning the bilevel optimization setup, managing multiple actors (one global, potentially many local), and ensuring stable training with the Reinforce algorithm likely requires significant expertise and effort.


2. The reported computational overhead, while quantified (~15%), is non-negligible and could be a barrier to practical adoption. This additional runtime cost for actor updates and reward computations might be prohibitive, especially for resource-intensive LLMs or scenarios requiring frequent retraining.


3. The method's performance relies heavily on the specific choices for reward functions and the difficulty grouping metric. While the chosen rewards (L2 gradient norm, perplexity ratio) and the SuperFiltering metric  show good results in the experiments, their universal optimality or robustness across different datasets, tasks, or model architectures is not guaranteed. The framework might be sensitive to these design choices.

4. The paper would be improved by comparing this RL-based actor approach to other recent dynamic strategies for multi-domain data, including those based on sample interactions[1] or alternative optimization objectives [2]

[1] Boosting Multi-Domain Fine-Tuning of Large Language Models through Evolving Interactions between Samples. ICML 2025.

[2] Mixture-of-Skills: Learning to Optimize Data Usage for Fine-Tuning Large Language Models. EMNLP 2024.

**Questions:**

1. Could the authors provide more specifics about the SuperFiltering metric  used to partition data into difficulty groups? How is difficulty defined and measured by this metric, and how robust is the local balancing performance if a different difficulty metric were used?

2. The cyclical patterns observed in the local sampling probabilities (e.g., Figure 2c) are interesting. Is this behavior consistently observed across different datasets and models? What mechanism within the HBO framework drives this apparent emergent curriculum?

3. Please review the formatting of Equation 8; the "where" clause defining PPL seems awkwardly placed on the same line and might be clearer if moved below or reformatted.

---

> ### Author Response · Authors · 2025-11-19
> **Response to Reviewer 9MNG (part1)**
>
> Thank you for your constructive feedback on our paper and for highlighting these important points. We appreciate the opportunity to clarify and strengthen our work. Below, we address each of your concerns in detail.
>
>
>
> > **Q1:** The framework introduces substantial complexity compared to standard fine-tuning or simpler dynamic sampling. Implementing and tuning the bilevel optimization setup, managing multiple actors (one global, potentially many local), and ensuring stable training with the Reinforce algorithm likely requires significant expertise and effort.
> > >
> > > **R1:** We appreciate the reviewer's concern about the framework's components. While HBO does integrate bilevel optimization, multiple actors, and a REINFORCE algorithm, we argue this structure is a necessary and well-justified solution to the complex, hierarchical nature of data imbalance, which simpler methods fail to address.
> > >
> > > Recent dynamic methods treat datasets as single blocks, such as Mixture-of-Skills, ignoring the rich internal heterogeneity (e.g., difficulty). To solve this, a hierarchical approach is required. The Global Actor manages the across-dataset imbalance, while the novel Local Actors manage the within-dataset imbalance.
> > >
> > > The design of the actors is lightweight, implementing them as simple 2-layer fully connected networks. This adds minimal architectural complexity.
> > >
> > > The use of REINFORCE algorithm is a standard and well-understood approach to optimize a non-differentiable sampling process, which is exactly what our problem requires
> > >
> > > We believe the modest increase in engineering is a worthwhile trade-off for the new capability (joint global-local balancing) and the significant performance gains demonstrated in Table 1. **Furthermore, we will release our code to facilitate adoption by the community.**
>
>
> > **Q2:** The reported computational overhead, while quantified (~15\%), is non-negligible and could be a barrier to practical adoption. This additional runtime cost for actor updates and reward computations might be prohibitive, especially for resource-intensive LLMs or scenarios requiring frequent retraining.
> > >
> > > **R2:** We thank the reviewer for focusing on practical adoption. We were transparent in quantifying this cost, reporting ~15\% additional training overhead compared to static sampling. We contend that this cost is non-prohibitive for two main reasons:
> > > - Favorable Trade-off: This 15\% overhead delivers consistent and significant performance gains across all three model backbones (e.g., +5.54 $\mu_{MT}$ on Llama-3.1-8B compared to static sampling). In the context of LLM fine-tuning, such accuracy improvements are often well worth a modest increase in training time.
> > > - Tunable Cost: The computational overhead is not fixed and can be adjusted based on user needs. As shown in our analysis in Figure 4 of Appendix B, this overhead is a function of the actor update frequency. More frequent updates can lead to higher overhead, while less frequent updates reduce it. Our chosen frequency (every 200 steps) provides the best balance of performance and efficiency, but a user could opt for less frequent updates (e.g., every 500 steps) to reduce the overhead further, while still achieving performance gains.

---

> ### Author Response · Authors · 2025-11-19
> **Response to Reviewer 9MNG (part2)**
>
> > **Q3:** The method's performance relies heavily on the specific choices for reward functions and the difficulty grouping metric. While the chosen rewards (L2 gradient norm, perplexity ratio) and the SuperFiltering metric show good results in the experiments, their universal optimality or robustness across different datasets, tasks, or model architectures is not guaranteed. The framework might be sensitive to these design choices.
> > >
> > > **R3:** This is an excellent point. We have conducted the extensive ablation on the reward functions in Appendix B. We investigate the impact of reward functions with comparing the combination of $L_2$ norm, CosSim, PPL ratio, PPL, loss, and present the results in Table 7.  We observe that all the combinations of reward function achieve consistent performance gains compared to the best heuristic baseline Prop (46.24) in $\mu_{ML}$, and the combination of $L_2$ norm and PPL Ratio achieves the best performance among all these combination. These findings validate the effectiveness and robustness of our framework. These results were placed in Appendix B due to space limitations, and we will relocate them to the main paper in the revision to highlight their importance.
> > >
> > > Besides these results demonstrating the robustness of our approach, we indeed acknowledge that these design choices may not be universally optimal. However, our experiments show that HBO is compatible with various reward functions. We believe this flexibility is a strength of the HBO framework, allowing it to be adapted to different scenarios as needed. As the field progresses, novel reward functions or difficulty metrics will definitely arise, and HBO can incorporate these advancements seamlessly.
>
> > **Q4:** The paper would be improved by comparing this RL-based actor approach to other recent dynamic strategies for multi-domain data, including those based on sample interactions[1] or alternative optimization objectives [2]
> > [1] Boosting Multi-Domain Fine-Tuning of Large Language Models through Evolving Interactions between Samples. ICML 2025.
> >
> > [2] Mixture-of-Skills: Learning to Optimize Data Usage for Fine-Tuning Large Language Models. EMNLP 2024.
> > >
> > > **R4:** We thank the reviewer for these relevant citations. For the paper 'Mixture-of-Skills' [2], we would like to clarify that this paper is one of our primary baselines. The "MoS" baseline , referenced throughout our paper (e.g., Table 1, Table 4, Figure 3).
> > >
> > > For the paper [1], as it is from ICML 2025, it appears to be concurrent with our own submission. As the code of this work is not open-sourced, we re-implement this baseline EVIC and conduct additional experiments in our multilingual setting with Llama-3.1-8b backbone. As shown in the following table, we observe that HBO outperforms EVIC across all the benchmarks.
> > >
> > >
> |   Metric    | $\mu_{ML}$ |   MMMLU   |    MGSM   |   XCOPA   |    XStoryCloze    |     XNLI     |
> |---|---|---|---|---|---|---|
> |         EVIC     |    46.10    |     41.76  |   19.73   |     62.70 |     63.95 |     42.36   |
> |         HBO   |      48.07  |     44.28 |     20.40 |     63.00 |     65.98 |     46.67    |

---

> ### Author Response · Authors · 2025-11-19
> **Response to Reviewer 9MNG (part3)**
>
> > **Q5:** Could the authors provide more specifics about the SuperFiltering metric used to partition data into difficulty groups? How is difficulty defined and measured by this metric, and how robust is the local balancing performance if a different difficulty metric were used?
> > >
> > > **R5:** Thank you for your valuable comments. This metric essentially is Instruction Following Difficulty (IFD) score [1] but computed using a smaller language model, which is Qwen2.5-0.5B in this work. SuperFiltering [2] demonstrates that a small language model is equally effective in computing IFD scores for data selection compared to using a large language model. IFD score is a pure statistical metric, that compares the losses or perplexities when the model generates a response $y_i$ with and without instructional context $x_i$, measuring how much help the instruction provides to the generation of the corresponding response. For a given instruction-following data pair, the IFD score is calculated as: $\texttt{IFD}(y_i|x_i) = \frac{\texttt{PPL}(y_i|x_i)}{\texttt{PPL}(y_i)}$, where $\texttt{PPL}(y_i|x_i)$ and $\texttt{PPL}(y_i)$ denote the perplexities of the given model in fitting response $y_i$ with and without the instruction $x_i$, respectively. A higher IFD score, indicating less instructional help, suggests a greater difficulty. On the contrary, the low IFD score represents that the given instruction can directly benefit the language model largely even without further training, representing the easiness and necessity of the instruction. For instance, if $x_i$ contains strong clues or direct answers, the model can easily generate $y_i$ with $x_i$, leading to a low IFD score.
> > >
> > > Regarding robustness, we conduct additional experiments with using PPL and loss as metrics. Here are the results with Llama-3.1-8b backbone:
> > >
> |   Metric    | $\mu_{ML}$ |   MMMLU   |    MGSM   |   XCOPA   |    XStoryCloze    |     XNLI     |
> |---|---|---|---|---|---|---|
> |         Prop.     |      46.25  |     40.78 |     18.00 |     62.70 |     64.63 |     45.11   |
> |         HBO-SuperFiltering     |      48.07  |     44.28 |     20.40 |     63.00 |     65.98 |     46.67    |
> |    HBO-PPL  |   47.42     |   44.71   |   19.33   |   62.60   |   64.65   |     45.81    |
> |     HBO-Loss     |   47.23     |   44.88   |   19.04   |   62.50   |   64.39   |     45.34    |
> > >
> > > Both of HBO-PPL and HBO-Loss achieve the performance gains compared to the best heuristic baseline Prop. It demonstrates that as long as the metric provides a meaningful signal of difficulty, the local actors will learn to balance them effectively.
> > >
> > > Reference:
> > > 1. Ming Li, Yong Zhang, Zhitao Li, Jiuhai Chen, Lichang Chen, Ning Cheng, Jianzong Wang, Tianyi Zhou, and Jing Xiao. 2024. From Quantity to Quality: Boosting LLM Performance with Self-Guided Data Selection for Instruction Tuning. In Proceedings of the 2024 Conference of the North American Chapter of the Association for Computational Linguistics: Human Language Technologies (Volume 1: Long Papers), pages 7602–7635, Mexico City, Mexico. Association for Computational Linguistics.
> > > 2. Ming Li, Yong Zhang, Shwai He, Zhitao Li, Hongyu Zhao, Jianzong Wang, Ning Cheng, and Tianyi Zhou. 2024. Superfiltering: Weak-to-Strong Data Filtering for Fast Instruction-Tuning. In Proceedings of the 62nd Annual Meeting of the Association for Computational Linguistics (Volume 1: Long Papers), pages 14255–14273, Bangkok, Thailand. Association for Computational Linguistics.

---

> ### Author Response · Authors · 2025-11-19
> **Response to Reviewer 9MNG (part4)**
>
> > **Q6:** The cyclical patterns observed in the local sampling probabilities (e.g., Figure 2c) are interesting. Is this behavior consistently observed across different datasets and models? What mechanism within the HBO framework drives this apparent emergent curriculum?
> > >
> > > **R6:** We are glad the reviewer found this observation interesting, as we believe it is a key insight. This dynamic behavior is consistently observed. While the exact shape and period of the cycle vary by dataset (compare the English-Local cycle in Fig 2c to the Math-Local dynamics in Fig 2d), the key finding is that the sampling probabilities are highly dynamic and non-static across all tested datasets
> > >
> > > This "emergent curriculum" is driven directly by the Local Reward function. The reward is a ratio of current perplexity to initial perplexity: $\mathcal{R}_{local} = PPL(\theta) / PPL(\theta_0)$.
> > > - Imagine the Local Actor focuses on "Hard" (Group 4) examples. The model learns, $PPL(\theta)$ for Group 4 decreases, and thus its $\mathcal{R}_{local}$ reward drops.
> > > - The REINFORCE algorithm then updates the actor's policy, reducing the sampling probability for Group 4 (which now has a low reward) and increasing it for other groups (e.g., "Easy" Group 1) whose relative reward is now higher.
> > > - This process repeats, causing the actor to "cycle" its attention to whichever group offers the most learning progress relative to its initial state.
> > > This behavior is not explicitly programmed; it is the optimal strategy learned by the Local Actor to maximize its cumulative reward.
>
> > **Q7:** Please review the formatting of Equation 8; the "where" clause defining PPL seems awkwardly placed on the same line and might be clearer if moved below or reformatted.
> > >
> > > **R7:** Thank you for this formatting suggestion. We will move the definition of $PPL(\cdot)$ to its own, separate line below the main equation in our future version to improve clarity.

---

### Official Review · Reviewer_dqCa · 2025-11-01

**Soundness:** 3
**Presentation:** 3
**Contribution:** 2
**Rating:** 4
**Confidence:** 3

**Summary:**

This paper introduces Hierarchical Balancing Optimization for fine-tuning large language models (LLMs) on diverse datasets by addressing both global and local data imbalance and heterogeneity. HBO uses a bilevel optimization framework with two types of actors: a Global Actor, which adjusts data allocation across different datasets, and Local Actors, which optimize data usage within individual datasets based on difficulty levels. Extensive experiments show that HBO consistently outperforms existing methods, offering significant improvements in model performance. The proposed method provides a comprehensive solution to the challenges of data imbalance and heterogeneity during fine-tuning.

**Strengths:**

1) HBO effectively addresses both global and local data imbalances, providing a more comprehensive solution to the challenges of fine-tuning LLMs on diverse datasets.

2) The bilevel optimization framework with Global and Local Actors allows for fine-grained control over data sampling, leading to improved model performance across various tasks.

3) Extensive experiments demonstrate HBO's strong applicability across multiple LLM backbones and tasks, consistently outperforming existing baselines and offering superior accuracy gains.

**Weaknesses:**

1) My main concern is the proposed method adds more computations based on MoS. The reinforcement learning framework, as well as some reward, is similar to the MoS method. This work adds more actors and the grad norm reward, more insights of this field could be added.

2) This paper primarily compares three sampling balancing methods: MoS, MultiUAT, and MultiDDS. However, many of the results are similar to uniform sampling. What is the next step of this field could be discussed.

**Questions:**

See weaknesses.

---

> ### Author Response · Authors · 2025-11-19
> **Response to Reviewer dqCa (part1)**
>
> Thank you for your thorough review of our paper and for the insightful questions you've raised. We appreciate the opportunity to address your concerns and clarify various aspects of our work. Below, we provide detailed responses to each of your points.
>
> > **Q1:** My main concern is the proposed method adds more computations based on MoS. The reinforcement learning framework, as well as some reward, is similar to the MoS method. This work adds more actors and the grad norm reward, more insights of this field could be added.
> > >
> > > **R1:**  Thank you for your valuable comments. We agree that our framework builds on the effective use of reinforcement learning for data sampling. However, we must clarify the fundamental novelty and contribution of HBO, which directly addresses a critical limitation of MoS and other existing methods.
> > >
> > > 1. Hierarchical vs. Flat Optimization
> > >
> > > The central limitation of prior dynamic methods like MoS is that they operate only at the global (across-dataset) level. They treat entire datasets as monolithic blocks, ignoring the vast imbalance and heterogeneity within each individual dataset.
> > >
> > > Our primary contribution is that HBO is the first to simultaneously optimize data allocation at two levels:
> > > - Global Actor: Balances sampling across different datasets (e.g., tasks or languages).
> > > - Local Actors: A novel component that balances sampling within each dataset based on internal characteristics like example difficulty.
> > > This hierarchical structure is not an incremental addition; it is a new approach designed to solve a problem that MoS and others overlook. Our ablation study in Table 2 directly validates this:
> > > - Removing the Local Actors (making HBO similar to a standard global-only method) causes a significant performance drop of 0.85 average performance.
> > > - Removing the Global Actor (relying only on local balancing) drops average performance by 0.61
> > > The full HBO model significantly outperforms both ablations, proving that both actors are essential and work in concert.
> > >
> > > 2. Reward Functions
> > >
> > > Our reward functions are specifically tailored to this hierarchical structure:
> > > - Global Reward ($L_2$ Norm): We use the gradient norm as it is an "ideal signal of learning dynamics"11. It effectively prioritizes datasets where the model "has more to learn," guiding the global balancing.
> > > - Local Reward (Perplexity Ratio): This reward is designed for the local task. It measures the relative improvement in performance on specific difficulty groups, allowing the model to dynamically shift focus between easy and hard examples within a dataset.
> > >
> > > 3. New Insights
> > >
> > > Our experiments provide new insights into the importance of addressing both global and local imbalance. We demonstrate the effectiveness of easy examples in Table 5 and illustrate a dynamic cyclical pattern in sampling distribution in Figure 2c. These findings bring new understanding and insights to the field.

---

> ### Author Response · Authors · 2025-11-19
> **Response to Reviewer dqCa (part2)**
>
> > **Q2:** This paper primarily compares three sampling balancing methods: MoS, MultiUAT, and MultiDDS. However, many of the results are similar to uniform sampling. What is the next step of this field could be discussed.
> > >
> > > **R2:** We thank the reviewer for this question, as it allows us to highlight the strength of our results. The paper compares HBO against seven baselines: three heuristic (Prop., Temp., Uni.) and four dynamic (MultiDDS, MultiUAT, MoS, and stronger MoS+).
> > >
> > > While some individual task scores in Table 1 may appear close, the claim that "many of the results are similar to uniform sampling" is not supported by the overall macro-average scores, which are the primary metric for these multi-dataset setups.
> > >
> > > **HBO consistently and significantly outperforms Uniform sampling:**
> > > - Llama-3.1-8B (Multitask): HBO (52.28) vs. Uniform (46.75) — a +5.53 gain.
> > > - Llama-3.1-8B (Multilingual): HBO (48.07) vs. Uniform (44.38) — a +3.69 gain.
> > > - Qwen2.5-7B (Multitask): HBO (60.37) vs. Uniform (57.42) — a +2.95 gain.
> > > - Qwen2.5-7B (Multilingual): HBO (55.21) vs. Uniform (53.90) — a +1.31 gain.
> > > - EuroLLM-9B (Multitask): HBO (50.16) vs. Uniform (48.07) — a +2.09 gain.
> > > - EuroLLM-9B (Multilingual): HBO (49.37) vs. Uniform (47.49) — a +1.88 gain.
> > >
> > > In every single setup of Table 1, HBO achieves the best overall performance, demonstrating a clear advantage over all heuristic and dynamic baselines.
> > >
> > > Furthermore, we conduct additional bootstrap-based statistical significance tests [1] and confirm that **these gains over the best baseline approach are statistically significant (p < 0.05) in terms of $\mu_{ML}$ and $\mu_{MT}$.**
> > >
> > > As for the next steps in this field, we believe proposing a unified framework that combines data rebalancing, data selection, and data synthesis would be a promising direction. Our HBO framework and other dynamic rebalancing methods, like MoS, focus on rebalancing existing data. There is also a line of work on data selection and data synthesis. These approaches are commonly studied independently. A unified framework that jointly optimizes rebalancing, selection, and synthesis could yield complementary benefits, leading to even more effective fine-tuning strategies for LLMs.
> > >
> > > We will include these additional results and discussions in our revision to strengthen this work.
> > >
> > > References:
> > > 1. Philipp Koehn. 2004. Statistical Significance Tests for Machine Translation Evaluation. In Proceedings of the 2004 Conference on Empirical Methods in Natural Language Processing, pages 388–395, Barcelona, Spain. Association for Computational Linguistics.

---

### Meta-Review · Area_Chair_b6Up · 2026-01-07

**Summary:**

The idea is straightforward but well-targeted. It addresses imbalance both across datasets and within each dataset. Experiments are broad and results are consistently positive; the revision also strengthens the evidence. The main trade-off is extra complexity&overhead, but it looks acceptable. Thus I recommend acceptance.

**Reviewer Concerns:**

dqCa: Feels too close to prior dynamic sampling work and the added RL machinery is not clearly worth the cost.

9MNG: Wants stronger evidence of robustness and clearer cost benefit given the added training overhead.

9Qbd: Main concerns were missing baselines and unclear evaluation, largely addressed in the revision and rebuttal.

9Boj: Questions practicality and scalability, and whether the reported improvements justify the complexity.

**Reviewer Scores:**

Reviewer dqCa likely stays at 4, at most 5
Reviewer 9MNG likely stays at 6, small chance of 7
Reviewer 9Qbd likely moves from 4 to 6
Reviewer 9Boj likely stays at 6, small chance of 7

---

### Decision · Program_Chairs · 2026-01-26

Accept (Poster)